# Methods of investigating forecast error sensitivity to ensemble size in a limited-area convection-permitting ensemble

Ross Noel Bannister<sup>1</sup>, Stefano Migliorini<sup>1,2</sup>, Alison Clare Rudd<sup>1,3</sup>, and Laura Hart Baker<sup>1</sup>

<sup>1</sup>Dept. of Meteorology, Univ. of Reading, Earley Gate, RG6 6BB, UK.

<sup>2</sup>Now at Met Office, FitzRoy Road, Exeter, EX1 3PB, UK.

<sup>3</sup>Now at Centre for Ecology and Hydrology, Wallingford, OX10 8BB, UK.

Correspondence to: Ross Bannister (r.n.bannister@reading.ac.uk)

**Abstract.** Ensemble-based predictions are increasingly used as an aid to weather forecasting and to data assimilation, where the aim is to capture the range of possible outcomes consistent with the underlying uncertainties. Constraints on computing resources mean that ensembles have a relatively small size, which can lead to an incomplete range of possible outcomes, and to inherent sampling errors. This paper discusses how an existing ensemble can be relatively easily increased in size,

5 it develops a range of standard and extended diagnostics to help determine whether a given ensemble is 'large enough' to be useful for forecasting and data assimilation purposes, and it applies the diagnostics to a convective-scale case study for illustration. Diagnostics include the effect of ensemble size on various aspects of rainfall forecasts, kinetic energy spectra, and (co)-variance statistics in the spatial and spectral domains.

The work here extends the Met Office's 24 ensemble members to 93. It is found that the extra members do develop a sig-10 nificant degree of linear independence, they increase the ensemble spread (although with caveats to do with non-Gaussianity), they reduce sampling error in many statistical quantities (namely variances, correlations, and length-scales), and improve the effective spatial resolution of the ensemble. The extra members though do not improve the probabilistic rain rate forecasts.

It is assumed that the 93-member ensemble approximates the error-free statistics, which is a practical assumption, but the data suggests that this number of members is ultimately not enough to justify this assumption, and therefore more ensembles are

15 likely required for such convective-scale systems to further reduce sampling errors, especially for ensemble data assimilation purposes.

# Copyright statement.

# 1 Introduction

Many operational centres use ensemble prediction systems (EPSs) to enhance the value of deterministic forecasts. An EPS allows a subset of possible alternative forecast outcomes to be assessed, and for aspects of the probability density function (PDF) of forecast uncertainty (e.g. its spread) to be estimated. These activities are useful for purposes such as forecast evaluation and data assimilation (DA). Only a relatively small number of ensemble members is affordable though and it is well known that this

can lead to shortcomings in sampling the probability density function (PDF) of forecast errors, especially in high-dimensional systems, like those used in numerical weather prediction.

Even when the PDF of forecast errors is truly Gaussian – in which case it is determined by its mean and covariance matrix – the sampled forecast error covariance matrix can represent forecast uncertainty along only a small number of directions in phase

- space, see e.g. Houtekamer and Zhang (2016). This issue can lead to the ensemble underestimating the true spread and it can 5 introduce noise in the sample covariances, which is particularly evident between distant points. The ensemble data assimilation community is acutely aware of this problem, see e.g. Houtekamer and Mitchell (1998); Hamill et al. (2001); Ehrendorfer (2007); Houtekamer and Zhang (2016); Bannister (2017), where methods of mitigating sampling error have been developed such as localisation, ensemble inflation, and hybridisation. For some applications of ensemble prediction, having more than
- a few tens of members does not provide added benefit (Talagrand et al., 1997; Houtekamer et al., 2014), but this is not true 10 in general. Houtekamer and Mitchell (2005) and Kondo and Miyoshi (2016) in particular argue that at least 10000 ensemble members would be needed in ensemble DA to avoid using the mitigation techniques mentioned above, but this is currently an impractical proposition for operational purposes. There are additional problems at convective-scales, where forecast errors are expected to have a significant degree of non-Gaussianity. At small horizontal scales the Rossby number is not small and the
- 15 vertical and horizontal scales of motion are comparable, which means that the equations describing the evolution of the flow contain significant non-linear terms (e.g. the velocity advection terms) (Zhang, 2005). A small ensemble is therefore also likely to fail to capture adequately non-Gaussian effects such as multi-modality of PDFs. Large ensembles would also be required to capture rare events, but this aspect is not the focus of our study.

Despite these problems with sampling error in practical ensemble sizes, operational weather centres are making increasing

- use of convection-permitting EPSs (e.g. with the AROME model (Seity et al., 2011; Bouttier et al., 2016), the COSMO model 20 (Gebhardt et al., 2011; Ben Bouallègue and Theis, 2014; Harnisch and Keil, 2015; Bick et al., 2016; Schraff et al., 2016), the Met UM (Bowler et al., 2008; Golding et al., 2014; Tennant, 2015), and the WRF model (Schwartz et al., 2014; Luo and Chen, 2015; Schwartz et al.,  $2015^{1}$ ) with the aim of producing skillful forecasts, including forecasts of convective precipitation whose predictability can vary with time. For the reasons mentioned above, it is useful to have a range of diagnostics available to help determine how many ensemble members are required to provide a sufficiently accurate characterization of a forecast 25

#### 1.1 Sensitivity to ensemble size in large-scale systems

error PDF for forecasting (including nowcasting) and data assimilation purposes.

The problem of determining the forecast sensitivity to ensemble size has been investigated in large-scale systems for a number of years. Buizza and Palmer (1998) considered ensembles of 2, 4, 8, 16 and 32 members in the ECMWF system and showed

30

that the skill of the 500 hPa geopotential height forecast increased with ensemble size, but depended on the specific forecast error norm used in a given verification method. In a related ECMWF study (Buizza et al., 1998) the benefits achieved by increasing the ensemble size versus increasing the model's resolution were studied. They considered ensembles of 32, 50

<sup>&</sup>lt;sup>1</sup>AROME = Application de la Recherche à l'Opérationnel à Méso-Echelle, COSMO = Consortium for Small-scale MOdeling, UM = Unified Model, WRF = Weather Research and Forecasting model.

and 128 members of different resolutions over 14 case studies. They showed that although increases in both ensemble size and resolution are beneficial, larger ensembles achieve a better skill. In particular they found better probability predictions of temperature and precipitation, as measured by the Brier score. Higher resolution forecasts did increase the ensemble spread, but insufficiently. Later Mullen and Buizza (2002) used the ECMWF's EPS to show that increasing only the resolution does

- 5 provide clear benefits in precipitation forecast skill of lighter rain, but increasing the number of members provides better value for forecasts of heavier rain (as estimated from a simple cost-loss model). More recently Bonavita et al. (2011) used the ECMWF's ensemble of data assimilations (EDA) to show that the sample forecast error standard deviation of the vorticity fields calculated with 10 and 50 member ensembles are highly correlated (between 80% and 95%), despite the ensembles having a larger sampling noise at smaller scales. In order to reduce sampling noise, forecast error variances need to be cleaned up, which
- 10 was done in spectral space using a filter with a truncation wave number estimated from climatological statistics. Forecast error standard deviations estimated with larger ensembles tend to need less filtering and therefore tend to have a higher effective spatial resolution, leading to better forecast scores.

### 1.2 Sensitivity to ensemble size in convective-scale systems

The conclusions of the above studies do not necessarily apply to EPSs using a convection-permitting ensemble, given the highly varying nature of predictability at convective-scales and their potential deviation from Gaussianity. For these reasons, work has also been done to study the effect of changing the number of ensemble members in convection-permitting model forecasts and some important studies are mentioned below.

Tong and Xue (2005) performed a synthetic observation study, assimilating perfectly modelled synthetic radar Doppler and reflectivity observations with an EnKF into a 2 km grid-length configuration of the ARPS<sup>2</sup> model. Although they used 100

- 20 members, they commented that as few as 40 members were enough to produce good analyses. Clark et al. (2011) used a 4 km grid-length configuration of WRF over the central United States. They took between 1 and 17 members over multiple cases to show that the area under the ROC curve for 6-h accumulated precipitation at various thresholds increased with increasing ensemble size for all considered scales. They found that there was little impact on the ROC area by increasing the ensemble size above 9 members, although they postulated that at least 60 members would be needed to bring sampling error and under
- 25 dispersiveness down to acceptable values. They also argued that more members are required for skillful forecasts of rare events or in low-predictability regimes. Bouallegue et al. (2013) found an improvement in the reliability and resolution of precipitation ensemble forecasts by increasing the number of members from 20 to 60 in a 2.8 km grid-length version of the COSMO-DE model. Ménétrier et al. (2014) discussed the characteristics of the forecast error variances and correlation length scales for small (6 member) and large (84 member) ensembles of forecasts using the convection-permitting AROME model (2.5 km
- 30 grid-length) over France, and focused on a case study characterized by strong convection. They investigated the effects of sampling errors in the small and large ensembles by comparing the forecast error standard deviations of near surface specific humidity derived from each. The small ensemble showed larger variability at small scales in particular, due to larger sampling noise. They calculated anomaly correlations between variance maps computed from the two ensembles for a range of quantities

<sup>&</sup>lt;sup>2</sup>Advanced Regional Prediction System.

and found values between  $\sim 60\%$  and 80%. They also showed that the correlation length-scales for a range of quantities are, on average, shortened as the number of ensemble members is increased (which is consistent with the need for less localisation for larger ensembles). In the case (at least) of specific humidity fields though, their study found that this effect is dominated by a decrease in the number of instances of long correlation length-scales in their large ensemble compared to their small

- 5 ensemble. The number of instances of small correlation length-scales in their large ensemble though was found actually to increase (their Fig. 13), thus complicating the effect that sampling error has on the length-scales. Schwartz et al. (2014) studied ensemble sizes of 5, 10, 20, 30, 40, and 50 members in a 3 km grid-length version of WRF over the central United States. They found that there were sometimes clear improvements in reliability, fractions skill score, and area under the ROC curve for precipitation forecasts. This was especially true for low precipitation thresholds, higher probability events, and longer forecasts.
- 10 Their conclusion though was that as few as 20 to 30 members showed similar skill to the full 50 members in many respects. Harnisch and Keil (2015) found increases in Brier skill score for precipitation forecasts, and decreases in the CRPS for 10 m winds, and in the average forecast-minus-observation departures of various quantities, by increasing the number of members from 20 to 60 in a 2.8 km grid-length version of the COSMO-DE model. These benefits were found to be significantly larger than using various ensemble inflation methods, or the introduction of a model error scheme.
- The overall conclusion from these studies is that the ensembles give improved forecast skill as the ensemble size is increased, but a judgment of the number of ensemble members required to achieve a particular goal depends on the model and on the application. Running high-resolution forecasts operationally remains an expensive activity, and so any studies indicating the degree of sensitivity of a range of diagnostics to ensemble size is valuable, especially when applied to models and outputs that have not been studied in this way, such as to convective-scale model forecasts of rainfall rate as is done in this paper.

# 20 1.3 Aims and scope of this paper

Throughout this work the authors had resources to generate a large ensemble for only a single convective-scale case study. This paper is intended to do four things: (i) highlight some issues around sampling error in convective-scale systems, (ii) document a means of generating a larger ensemble from an existing small ensemble, (iii) develop a number of potentially informative diagnostics, and (iv) test the diagnostics for the large ensemble. Due to the availability of only one case study, the results are therefore not intended to provide a definitive answer to the number of ensemble members required in the system studied, as to do this, more members, and more realisations would be required. It is hoped though that the methodology, the choice of diagnostics, and how they are interpreted will be useful to researchers and developers who do have the resources to use the tools to their full potential, especially those who are planning new or extended EPSs operationally.

A central assumption made in this work is that the large ensemble is sufficient to neglect sampling errors. Sub-samping from this large ensemble is then done to see how smaller ensembles (of varying size, now assumed to have sampling errors) reproduce aspects of the full ensemble. The methods are illustrated with a case study with an (up to) 93 member convectionpermitting forecast ensemble based on a 1.5 km grid-length version of the Met Office's Met UM over the Southern UK. We do acknowledge that 93 members is not sufficiently large to neglect sampling errors and, as stated above, and that a single case study is not sufficient to reach a definitive conclusion, but it is beyond the scope of our project resources to generate more

5

members or to study further case studies. The case study is meteorologically interesting though; it is characterized by multiple rain bands generated by a cold front passing over the model's domain. This paper attempts to use and develop methods to help answer the following kinds of questions that arise in ensemble studies.

- How can linearly independent extra members be generated from an existing ensemble?
- How can the ensemble size impact probabilistic forecasts of rainfall?
  - How can the ensemble size affect how the kinetic energy spectrum is resolved?
  - How can the ensemble size affect estimates of (co-)variability of thermodynamic and moisture fields?
  - Is the ensemble used in a particular application large enough to neglect sampling error?
- The structure of this paper is as follows. Section 2 is a description of the case study, Sect. 3 explains how the 93-member
  ensemble is created from the operational 24-member ensemble and examines how linearly independent the extra members are.
  The remaining sections describe how sensitive various diagnostics are to ensemble size. Section 4 looks at ensemble means and spreads, Sect. 5 looks at probabilistic aspects, Sect. 6 looks at kinetic energy spectra, and Sect. 7 looks at aspects of the forecast error (co)variances. Finally, Sect. 8 discusses the main conclusions.

# 2 The 20th September 2011 case study

- 15 This case study is of interest to the DIAMET<sup>3</sup> project. It comprises multiple cloud bands (labelled "1", "2", and "3" in Fig. 1) over Southern UK and it was intensively observed with a field campaign (Vaughan et al., 2015). The case is also studied by Baker et al. (2014) using a 24-member Ensemble Transform Kalman Filter (ETKF) ensemble mentioned in Sect. 3.4. The case is characterised by an eastward moving cold front over the southern UK as shown in Fig. 3 of Baker et al. (2014). Rainfall maps constructed from the UK's network of radar instruments are shown in Fig. 1 from 13Z (panel a) to 18Z (panel f). At 13Z there are three rain bands associated with the front, but only band 1 is within the domain at this time. Band 2 enters the domain at 14Z, and becomes very clear at 15Z and 16Z. Rain band 3 has just started to appear in the boundary of the domain at 16Z, and all three bands are within the domain at 17Z and 18Z. Bands 1 and 2 have started to capture. More details about this
  - case are given by Baker et al. (2014), and here we focus mainly on the ensemble at 15Z.

<sup>&</sup>lt;sup>3</sup>DIAbatic influences on Mesoscale structures in ExTratopical storms.