# Peer review of "Methods of investigating forecast error sensitivity to ensemble size in a limited-area convection-permitting ensemble"

_Geoscientific Model Development, 2017_

## Editor Comment (EC1) · J. D. Annan (Editor) · 20 Nov 2017

A couple of minor details that I perhaps should have asked for before initial publication, but which can hopefully be addressed in any revision:

We generally request that the paper title refer specifically to the model name/version number under consideration, i.e. in this case perhaps you could add to the title something like ":case study using model vX.X"

The code availability description is rather too vague. Please specify in more detail which pieces of code are available, and which if any are not (with some justification for

the latter).

However neither of these details need affect the peer review of rest of the content of the paper.

**[GMDD](https://doi.org/10.5194/gmd-2017-260)**

---

## Referee Comment (RC1) · Anonymous Referee #1 · 13 Dec 2017

**Review on "Methods of investigating forecast error sensitivity to ensemble size in a limited-area convection-permitting ensemble" by R. Bannister, S. Migliorini, A. C. Rudd and L. H. Baker.**

This paper examines the impact of ensemble size for the prediction of a particular rainy event over UK. In a first part some diagnostics on short-range forecast are examined for a small, an intermediate and a large ensemble. In a second part, the ability of these three ensembles to accurately estimate short-range forecast error variances and correlations is discussed.

Regarding the ensemble prediction part, this study, as mentioned by the authors, suffers from being based on a single case. Hence, neither objective scores nor robust conclusions can be derived, while we recall that a number of studies have already provided more comprehensive analyses of the ensemble size effect in convective-scale EPSs (Clark et al., 2011; Schwartz et al., 2017; Hagelin et al., 2017; Raynaud and Bouttier, 2017).

Regarding the estimation of error covariances, the authors mainly base their analysis on the sampling noise theory published in previous papers. Their additional developments are not clear to me and I suspect the methodology may be flawed to some extent. Overall, their results only confirm some well-known ideas. In particular, a similar comparison with a large 90-member ensemble is provided in Ménétrier et al. (2014).

In its present form, I thus consider the paper does not provide enough original and new results to warrant publication in GMD.

I provide hereafter a list of my major concerns.

**Main comments**

- 1. The generation of the large ensemble is not completely clear to me. Here are some points that require further details :

    — Figure 2 suggests that you negate analysis (instead of forecast) perturbations from the 23-member MOGREPS-G, could you confirm ? Are these (positive + negative) perturbations added to the 4D-Var control to form the 46-member MOGREPS-G ? Same question for the passage from 46 to 92 perturbed members.
    — The data assimilation step in MOGREPS-G adds ETKF perturbations around the control 4D-Var analysis. Is the ETKF size also increased for the DA cycling of the 46 and 92-member ensembles ?

- 2. Sections 7.2 and 7.3 : I don't fully agree with the application of the sampling noise theory to $\mathbf{g}^{(N)\prime}$ when $\mathbf{v}^{(\infty)} \approx \mathbf{v}^{(93)}$, and with the subsequent diagnostics.
In my opinion, one proper way to evaluate how suboptimal is a 93-member ensemble may be to derive an analytical expression for the covariance of the difference $\mathbf{v}^{(N)} - \mathbf{v}^{(93)}$ and to compare it to the 'true' sampling noise covariance. Then by varying the size of the large ensemble you could get an estimate of how large enough is the reference ensemble regarding different aspects of the covariance estimation. For instance, it is expected that an ensemble could be considered large enough for the variance estimate but not for the correlation.
Alternatively, note that the sampling noise theory indicates that the noise standard deviation of the 93-member ensemble is approximately $\sqrt{92/23} = 2$ times smaller than the noise standard deviation of the small 24-member ensemble.

- 3. Section 7 : Figures 11, 12, 15 and 16 do not provide any new results compared to previously published studies.

**Minor comments**

• Part 1.2 : in order to improve the understanding, please try to separate studies that concern ensemble forecasting from those that concern ensemble data assimilation since these are two different problems.

• How the four members of Figure 4 have been chosen ? Randomly ? A more relevant choice may be to show clusters from the large ensemble, since they may provide a better representation of the ensemble distribution.

• P11 L12 "This is consistent with the lack of sensitivity of ensemble mean forecast skill to ensemble size" : I don't agree with this sentence. Indeed, objective verification scores with convective-scale EPS clearly show an improvement of the ensemble mean skill when increasing the sample size, especially when starting from very small ensembles (e.g., 6 or 12 members). As you start from a larger ensemble this improvement is likely to be already close to saturation, which gives the feeling the ensemble mean skill is not sensitive to the sample size.

• Figures 4 and 5 : what is the rationale for showing the ETKF ensemble here ? It does not show the impact of ensemble size but rather the impact of data assimilation, which is not the subject of the paper.

• Figure 7 : could you explain how the frequency of each rank has been computed ? I don't understand why some ranks have a frequency equal to 1 while one would expect that all ranks frequencies sum to 1. In addition, it's quite unusual to overlay rank histograms of ensembles with different sizes (the x-axis should be different) and I find it very difficult to objectively compare the different ensembles on these figures. For instance, delta scores (distance to flatness) and number of outliers may provide a more understandable information.

• Section 7.5 : I don't understand why the correlation is estimated by dividing by $v_x^{(\infty)}$ and not $v_x^{(N)}$ ?

• P29 L30 "Not only does this show ... it is an interesting result in itself" : sampling properties of length-scale estimates have already been deeply documented in Pannekoucke et al. (2008); Raynaud and Pannekoucke (2012) for instance.

• At several places the authors say that it's difficult to judge if an ensemble is large enough to neglect sampling error. A possible useful tool is the signal-to-noise ratio (SNR), whose theoretical definition (e.g. Equation (17) of Ménétrier et al. (2015)) could be used to anticipate the SNR of the different components of the covariance matrix. In addition, estimating how large enough is the ensemble also depends on the subsequent analysis and forecasts scores.

• Final comment, you say that "the lack of sensitivity to other aspects, like rainfall properties and biases suggests that the quality of probabilistic forecasts would not be improved". This statement is in contradiction with objective scores from Hagelin et al. (2017), that indicate a clear improvement of precipitation forecasts when going from 12 to 24 members. In order to answer the question of the impact of ensemble size you need to look at relevant scores : ensemble mean, probabilities or rank histograms shown in this paper are clearly not sufficient and are known to saturate with 30-40 members. In order to better highlight the benefit of more members, it is recommended to look at scores that focus on the tails of the distribution, such as the weighted CRPS or quantile score for instance. In addition, the way the large ensemble is generated may also contribute to this lack of positive impact.

**Specific comments**

- P10 legend fig. 3 : to which pressure level does model levels 36 and 11 correspond ?

- P13 "The ensemble of rain rate .... a specified threshold rain rate" : please reformulate as this is not clear.

- P23 "The spectral weight of the variance sampling errors" : rather say this is the noise power spectrum.

**References**

Clark, A. J., J. S. Kain, D. J. Stensrud, M. Xue, F. Kong, M. C. Coniglio, K. W. Thomas, Y. Wang, K. Brewster, J. Gao, X. Wang, S. J. Weiss, and J. Du, 2011 : Probabilistic precipitation forecast skill as a function of ensemble size and spatial scale in a convection-allowing ensemble. *Monthly Weather Review*, **139**, 1410–1418.

Hagelin, S., J. Son, R. Swinbank, A. McCabe, N. Roberts, and W. Tennant, 2017 : The met office convective-scale ensemble MOGREPS-UK. *Quart. J. Roy. Meteor. Soc.*, **143**, 2846–2861.

Ménétrier, B., T. Montmerle, L. Berre, and Y. Michel, 2014 : Estimation and diagnosis of heterogeneous flow-dependent background-error covariances at the convective scale using either large or small ensembles. *Quart. J. Roy. Meteor. Soc.*, **140**, 2050–2061.

Ménétrier, B., T. Montmerle, Y. Michel, and L. Berre, 2015 : Linear filtering of sample covariances for Ensemble-Based Data Assimilation. Part II : Application to a Convective-Scale NWP Model. *Monthly Weather Review*, **143**, 1644–1664.

Pannekoucke, O., L. Berre, and G. Desroziers, 2008 : Background-error correlation length-scale estimates and their sampling statistics. *Quart. J. Roy. Meteor. Soc.*, **134**, 497–508.

Raynaud, L., and F. Bouttier, 2017 : The impact of horizontal resolution and ensemble size for convective-scale probabilistic forecasts. *Quart. J. Roy. Meteor. Soc.*

Raynaud, L., and O. Pannekoucke, 2012 : Sampling properties and spatial filtering of ensemble background-error length-scales. *Quart. J. Roy. Meteor. Soc.*

Schwartz, C. S., G. S. Romine, K. Fossell, R. Sobash, and M. Weisman, 2017 : Toward 1-km ensemble forecasts over large domains. *Monthly Weather Review*, **145**, 2943–2969.

---

## Referee Comment (RC2) · Anonymous Referee #2 · 12 Jan 2018

The comment was uploaded in the form of a supplement:
https://www.geosci-model-dev-discuss.net/gmd-2017-260/gmd-2017-260-RC2-supplement.pdf

---

## Author Comment (AC1) · 5 Feb 2018

**Responses to reviewers for, "Methods of investigating forecast error sensitivity to ensemble size in a limited-area convection-permitting ensemble" (gmd-2017-260)**

February 2018

This document is our general (undetailed) response to the main criticisms of reviewer 1. A second (more detailed) part will follow should the editor allow the paper to be revised with a reasonable chance of acceptance in GMD.

**Response to reviewer 1**

We would like to thank reviewer 1 for his/her comments and criticism of our manuscript sent to GMD.

We are disappointed and puzzled to see the reviewer's opinion on this work, especially his/her belief there is not enough original work in the paper (para. 4 of reviewer 1's report), and their suspicion that some of the analysis is flawed (para. 3). We hope to make a case that the paper does actually have significant and interesting new results. Of course virtually all scientific work has limitations, and we acknowledge ours, but we strongly believe that the work presented is useful and has enough new content, as

explained below. Unfortunately the reviewer's comments appear to give only a one-sided view.

1. Single case study. One of the comments concerns the presence of a only single case study, and this limitation is indeed highlighted by the authors (para. 2 of reviewer's report). It does mean that the specific results do not necessarily represent firm conclusions, but it certainly does not mean that the results are not useful, e.g. they are probably representative of the particular weather regime studied (especially as the case comprised several days' data). Menetrier et al. (2014), which the reviewer cites (para. 3), is also based on a single case, demonstrating that this need not be a show-stopper. The 'single case' limitation (which was outside of our control) is the reason why we have emphasised the methodology, rather than the specific results, and to seek publication in this particular journal. Indeed, parts of the methodology could be adopted by others studying other systems.

2. Originality. We believe that the reviewer's opinion on originality is not a fair judgment. The new aspects of the paper are not highlighted in the reviewer's report, even though they are present in the paper. A version of this manuscript was originally sent to another leading journal, and the only reason why it was not accepted there was because of the single case study limitation, which should not be a problem given the scope of GMD. Importantly in that submission, both reviewers' comments were otherwise very positive about the work (it was described as "state of the art", and a "significant contribution to the field"). Of course, that was the outcome of a separate editorial process, but it does serve to highlight that it is possible, as in the present case, to get a distorted view of a piece of work from a small number of reviewers. It is not clear to us why the present reviewer should hold their opinion, but it might be the case that he/she was expecting that every figure should represent a brand new diagnostic. It is usual for a study to use a range of new *and* standard diagnostics as part of an overall picture. Many of the

figures in the paper are in fact, we believe, based on new developments in the field of ensemble forecasting and/or data assimilation, which the reviewer has not discussed in his/her report, and this may wrongly lead the editor to believe that the paper contains no or little original content. To emphasise our point, these are areas of the manuscript that we think especially have not been explored before in the context of ensemble forecasting/DA:

(a) The way that a large ensemble can be generated from an existing smaller ensemble (Fig. 2, Sect. 3.2). This technique will almost certainly be of interest to other research groups who would like to extend their ensemble systems.

(b) The linear independence tests to show that the members do develop linear independence (Fig. 3, Sect. 3.3). This is a simple but essential diagnostic to confirm the usefulness of the above method.

(c) Study of the kinetic spectrum in an ensemble context (Figs. 9 and 10, Sect. 6). This suggests how errors in kinetic energy of a finite ensemble change as a function of scale – very important information to have when designing and interpreting ensemble data assimilation systems.

(d) Study of the form of the correlation functions of variance errors, in particular finding an excellent fit to an exponential form (Figs. 13 and 14, Sect. 7.3), and how this could be used to generate variance fields that have a prescribed form of sampling error characteristic of a finite ensemble (Eq. 7). The exponential fit makes the length-scale analysis different from that covered previously, e.g. Menetrier et al. (2014), Pannekoucke et al. (2008), and Raynaud and Pannekoucke (2012/3), which looked at parabolic or Gaussian forms (7th minor comment or reviewer's report).

(e) A potential new test of whether an ensemble is large enough to meaningfully neglect sampling error (Sect. 7.2, and Fig. 13).

(f) Analysis of the errors in the sub-sampling for many diagnostics (e.g. the fit to the exponential, Fig. 14).

(g) The application of the above and standard diagnostics to the high-resolution Met Office Unified Model.

3. Validity. The reviewer has questioned the validity of the assumption made in the paper that the large (93 member) ensemble has negligible sampling error (main comment 2). This is indeed an important issue, and is part of what makes this manuscript interesting scientifically. We have done what is usual in science: we have made an assumption, used it in an attempt to learn about a system, and then discussed how valid the assumption is. In our particular application there is evidence that this assumption may not be met (see discussion after Eq. 6), but that is still very useful information. This would be an unfair basis on which to reject the manuscript.

The reviewer's report contains some minor and detailed aspects which we can address in a way to improve the paper, should the manuscript be taken further.

---

## Author Comment (AC2) · 5 Feb 2018

**Responses to reviewers for, "Methods of investigating forecast error sensitivity to ensemble size in a limited-area convection-permitting ensemble" (gmd-2017-260)**

February 2018

This document is our general (undetailed) response to the main criticisms of reviewer 2. A second (more detailed) part will follow should the editor allow the paper to be revised with a reasonable chance of acceptance in GMD.

**Response to reviewer 2**

We would like to thank reviewer 2 for his/her comments and criticism of our manuscript sent to GMD.

The reviewer's report contains many positive comments about the paper, and in fact does show that there are many original and useful aspects to the work presented, despite the apparently negative overall score given.

Some of the points of reviewer 2 overlap with those of reviewer 1, namely the single case study (para. 2 of reviewer 2's report), and the apparent lack of originality (para.

3). We defend these issues in the response to reviewer 1 (reproduced below). We respond to other of the main issues raised by reviewer 2 as follows.

1. Content and focus. We accept that the manuscript could be shortened (see para. 4 of reviewer's report) and that the focus could be improved (paras. 3 and 5). The focus could be improved by covering how each issue is important to ensemble forecasting or DA. That said though, the two disciplines (ensemble forecasting and DA) are inherently linked, and could/should become aspects of the same problem in time, and so it is arguable that papers that are relevant to both should be welcomed. The extra references could be added (para. 6).

2. Scientific interpretation. We could reduce the interpretation in the light of the single case limitation (para. 7), although we think that there is still scope for interpretation as the results are still valid, although not necessarily applicable to all weather situations.

3. More points on novelty. In addition to our defense of originality covered in the response to reviewer 1, reviewer 2 does actually acknowledge that many of the results are indeed new and interesting (paras. 8, 9, 12), although he/she has chosen to make emphasis on other parts of the paper when making his/her overall assessment (as stated in our response to reviewer 1, our paper is a mixture of new and standard diagnostics to help paint an overall picture).

   (a) With respect to the comments about Sect. 7 (para. 12 of reviewer's report) about the assumed overlap with Menetrier et al. (2014): please note that there are some profound differences between our work and theirs, namely the (robust) finding of exponentially-shaped correlation functions with our work.

   (b) With respect to the comments about the kinetic spectrum (para. 11): we acknowledge that the mere computation of a kinetic spectrum is not new, but

its study in an ensemble context, we believe, is new. The reviewer questions why doing an average over ensemble members is necessary. In the same way that performing an ensemble average of a field can and does affect the size of features (as we reminded readers in Sect. 4.1 in the context of rainfall), this can also be true when looking at the effective resolution of an ensemble vs an individual member. (We could make more of this in a revision.)

4. Simplicity. The reviewer commended the work of Sect. 3.3 (linear independence tests), but commented that the method is complicated (para. 9 of reviewer's report). The method used is actually extremely simple (much simpler than, e.g. rank histogram computations in our opinion). The editor or reviewer is invited to see the code, which is made available with the paper.

The reviewer's report contains some minor and detailed aspects which we can address in a way to improve the paper, should the manuscript be taken further. We also hope that the editor will see that the overall assessment of this manuscript is not justified (e.g. that the presentation is actually much better than the "fair" assessment given).

**Relevant extract from report for reviewer 1**

1. Single case study. One of the comments concerns the presence of a only single case study, and this limitation is indeed highlighted by the authors (para. 2 of reviewer's report). It does mean that the specific results do not necessarily represent firm conclusions, but it certainly does not mean that the results are not useful, e.g. they are probably representative of the particular weather regime studied (especially as the case comprised several days' data). Menetrier et al. (2014), which the reviewer cites (para. 3), is also based on a single case, demonstrating that this need not be a show-stopper. The 'single case' limitation (which

was outside of our control) is the reason why we have emphasised the methodology, rather than the specific results, and to seek publication in this particular journal. Indeed, parts of the methodology could be adopted by others studying other systems.

2. Originality. We believe that the reviewer's opinion on originality is not a fair judgment. The new aspects of the paper are not highlighted in the reviewer's report, even though they are present in the paper. A version of this manuscript was originally sent to another leading journal, and the only reason why it was not accepted there was because of the single case study limitation, which should not be a problem given the scope of GMD. Importantly in that submission, both reviewers' comments were otherwise very positive about the work (it was described as "state of the art", and a "significant contribution to the field"). Of course, that was the outcome of a separate editorial process, but it does serve to highlight that it is possible, as in the present case, to get a distorted view of a piece of work from a small number of reviewers. It is not clear to us why the present reviewer should hold their opinion, but it might be the case that he/she was expecting that every figure should represent a brand new diagnostic. It is usual for a study to use a range of new *and* standard diagnostics as part of an overall picture. Many of the figures in the paper are in fact, we believe, based on new developments in the field of ensemble forecasting and/or data assimilation, which the reviewer has not discussed in his/her report, and this may wrongly lead the editor to believe that the paper contains no or little original content. To emphasise our point, these are areas of the manuscript that we think especially have not been explored before in the context of ensemble forecasting/DA:

   (a) The way that a large ensemble can be generated from an existing smaller ensemble (Fig. 2, Sect. 3.2). This technique will almost certainly be of interest to other research groups who would like to extend their ensemble systems.

   (b) The linear independence tests to show that the members do develop linear independence (Fig. 3, Sect. 3.3). This is a simple but essential diagnostic to confirm the usefulness of the above method.

   (c) Study of the kinetic spectrum in an ensemble context (Figs. 9 and 10, Sect. 6). This suggests how errors in kinetic energy of a finite ensemble change as a function of scale – very important information to have when designing and interpreting ensemble data assimilation systems.

   (d) Study of the form of the correlation functions of variance errors, in particular finding an excellent fit to an exponential form (Figs. 13 and 14, Sect. 7.3), and how this could be used to generate variance fields that have a prescribed form of sampling error characteristic of a finite ensemble (Eq. 7). The exponential fit makes the length-scale analysis different from that covered previously, e.g. Menetrier et al. (2014), Pannekoucke et al. (2008), and Raynaud and Pannekoucke (2012/3), which looked at parabolic or Gaussian forms (7th minor comment or reviewer's report).

   (e) A potential new test of whether an ensemble is large enough to meaningfully neglect sampling error (Sect. 7.2, and Fig. 13).

   (f) Analysis of the errors in the sub-sampling for many diagnostics (e.g. the fit to the exponential, Fig. 14).

   (g) The application of the above and standard diagnostics to the high-resolution Met Office Unified Model.

---

## Author Comment (AC3) · 5 Feb 2018

Dear Prof. Annan,

Thank you for your comments. I hope that we can address your minor issues should we have an opportunity to revise the manuscript with a good chance of acceptance.

You will have seen the recommendations of both reviewers. We strongly believe that these comments are unjustified, and we are very puzzled by them, as they do not seem to match the qualities of the manuscript itself. We hope that you will see for yourself that they are unfair assessments. Our responses are outlined in our reply to each of

the reviewers.

Many thanks.

---

## Editor Comment (EC2) · J. D. Annan (Editor) · 5 Mar 2018

Editor's comments.

Firstly, I would like to apologise for the time this has taken me.

Two reviewers have presented thorough and detailed reviews which include somewhat similar criticisms, primarily being that the manuscript does not offer much that is new, and also that the focus is rather confused between data assimilation and ensemble forecast assessment. They also offer a number of more detailed critical comments on the methodology. While the authors may feel they can rebut or take account of these,

the problem of content would remain. I don't see any compelling reason to overturn their judgements and therefore cannot recommend revision of the manuscript.